# Ultrasound-Based Assessment of Subcutaneous Adipose Tissue Changes During a 7-Day Ultramarathon: Association with Anthropometric Indices, Not Body Mass

**DOI:** 10.3390/jfmk10040467

**Published:** 2025-12-01

**Authors:** Daniela Chlíbková, Beat Knechtle, Katja Weiss, Ingrid Kováčová, Thomas Rosemann

**Affiliations:** 1Centre of Sports Activities, Brno University of Technology, 616 69 Brno, Czech Republic; daniela.chlibkova@vut.cz; 2Institute of Primary Care, University of Zurich, 8091 Zurich, Switzerland; 3Gesundheitszentrum St. Gallen, 9000 St. Gallen, Switzerland; katja@weiss.co.com (K.W.); thomas.rosemann@usz.ch (T.R.); 4Central European Institute of Technology, Masaryk University, 601 77 Brno, Czech Republic; kovacova.inge@gmail.com

**Keywords:** endurance performance, anthropometry, subcutaneous adipose tissue, ultrasound assessment, multistage ultramarathon, energy balance

## Abstract

**Background**: Accurately tracking body-composition changes in endurance field settings remains methodologically challenging. This study aimed to evaluate whether changes in subcutaneous adipose tissue (SAT) across a 7-day ultramarathon are better reflected by anthropometric indices than by body mass (BM) alone. **Methods**: Twenty ultrarunners were assessed using both anthropometric indices and ultrasound measurements of SAT thickness, applying a novel method that distinguishes layers including (D_I_) versus excluding (D_E_) embedded fibrous structures. Measurements were obtained before the race and after Stages 4 and 7. Indices included body mass index (BMI), mass index (MI_I_), and waist-to-height ratio (WHtR). **Results**: Total SAT thickness decreased significantly for both D_I_ (*p* = 0.001) and D_E_ (*p* < 0.001). BM, BMI, MI_I_, and WHtR also declined significantly post-race (*p* < 0.001). SAT reduction was most pronounced at the abdominal and thigh sites. Additionally, ultrarunners with lower D_E_ values exhibited lower fat at the abdomen and distal triceps. BMI was significantly related to D_E_ at the upper and lower abdomen and erector spinae; MI_I_ was significantly associated with D_E_ at the upper and lower abdomen; and WHtR correlated with both D_E_ and D_I_ at abdominal and erector spinae sites. BM showed no significant association with any SAT parameter. **Conclusions**: Ultrasound-derived SAT thickness, in combination with BMI, MI_I_, and WHtR, offers a field-feasible approach to evaluate body-composition change during multistage ultramarathons. In contrast, BM alone does not reliably reflect SAT distribution or loss.

## 1. Introduction

Accurately attributing changes in body composition, particularly during endurance performance and in relation to hydration status, remains a major methodological challenge, especially in field-based research. The more precisely body components are partitioned as body mass (BM) changes, the more reliable the inferences from real-world endurance events. Most studies on endurance performance still rely primarily on basic parameters such as BM, with some incorporating anthropometric indices. The most widely used, body mass index (BMI), estimates relative BM but does not account for body dimensions, tends to overestimate fat in muscular individuals, and fails to reflect fat distribution [1,2]. An improved mass index (MI_I_) proposed by W. Müller [3,4] incorporates sitting height relative to leg length. Other commonly used anthropometric indicators include the waist-to-hip ratio (WHR), a marker of abdominal obesity [5], and the waist-to-height ratio (WHtR), which better reflects fat distribution [1,6]. Although several studies infer changes in body fat, muscle mass, or water content from BM variation, fluid and electrolyte shifts during competition can substantially confound such assessments [7].

Furthermore, standardizing measurement conditions (pre-, mid-, and post-race) is rarely feasible in field settings. For example, glycogen supercompensation can increase water retention and alter skinfold compressibility [7]. Even widely used techniques such as skinfold calipers are susceptible to compression artifacts and operator error [8]. Although dual-energy X-ray absorptiometry (DXA) is often treated as the gold standard for total body composition, fat estimates in athletic populations may be inaccurate [7,9]. Potential sources of error include the compartmental algorithms used to separate muscle, fat, and bone and the population-based reference standards embedded in the analysis software [10]. Bioelectrical impedance analysis (BIA), commonly used in field settings, is likewise affected by multiple environmental and physiological factors that can diminish accuracy [11].

Ultrasound has emerged as a noninvasive, reproducible alternative for site-specific SAT assessment under real-world race conditions. Recent advances permit differentiation of fat layers that include versus exclude embedded fibrous structures, offering insight into microstructural adipose adaptations during extreme exercise. Accordingly, this study examined whether ultrasound-derived SAT changes over a 7-day ultramarathon align more closely with anthropometric indices (BMI, MI_I_, WHR, WHtR) than with body mass (BM). We hypothesized that anthropometric indices would better capture exercise-induced SAT variation than BM alone, supporting their practical use for field monitoring of endurance athletes.

## 2. Materials and Methods

### 2.1. Participants and Race

The study complied with the Declaration of Helsinki and was approved by the Ethics Committee of the Centre of Sports Activities, Brno University of Technology, Czech Republic (Approval No. 1/2016; 16 May 2016). All race entrants were invited to participate. All race participants were invited to join the study. Study information was distributed by the race organizers via email approximately two months and again one week before the event. Participation was voluntary, and no specific inclusion or exclusion criteria were applied. Twenty amateur ultrarunners (17 men, 3 women) were randomly selected from 21 volunteers who registered for the study; all 20 completed the entire race and were included in the analyses. Written informed consent for anonymous data use was obtained from all participants.

The study examined the Moravian Ultra Marathon 2016, an international multistage running race widely regarded as among the most demanding of its kind in the Czech Republic. The race was held in Lomnice, Czech Republic, from 3–9 July 2016 and comprised seven consecutive marathons (~42.2–43.0 km per stage) on seven successive days. Each stage followed a distinct point-to-point course over mixed terrain (asphalt with light traffic, forest paths, and field trails), with an average daily elevation gain of ~900 m, thereby imposing a substantially higher load than typical single-stage events. Six on-course aid stations per stage provided water, sports drinks, tea, soup, caffeinated beverages, mineral water, fruit, cheese, biscuits, crisps, peanuts, and chocolate (as conditions permitted). Athletes were allowed to use personal nutrition. Direct hydration markers were collected within the broader research program for this cohort but were not analyzed in the present study; accordingly, short-term BM changes attributable to fluid balance cannot be partitioned here by design.

### 2.2. Anthropometric Measurements

Anthropometric measurements included body mass (BM), standing height, sitting height, and waist and hip circumferences. Standing height was measured to the nearest 0.01 m with a portable anthropometer (A-226; Trystom, Olomouc, Czech Republic). Sitting height was recorded in a fully upright seated position; the height of the seat was subtracted to obtain true sitting height. BM was assessed to the nearest 0.1 kg using a calibrated digital scale (Beurer BF 15; Beurer GmbH, Ulm, Germany). Measurements were performed during race registration (1.5 h before Stage 1) and immediately after Stages 4 and 7. Participants wore standard running attire and were barefoot for all assessments. All measurements were obtained by a certified anthropometrist with extensive experience in ultrasound-based fat assessment. Body mass index (BMI) was calculated as BMI = BM/height^2^ (kg·m^−2^). The improved mass index (MI_I_), which incorporates sitting height, was calculated as MI_I_ = 0.53 × BM/(h × s), where h = standing height (m) and s = sitting height (m), following Müller et al. [3,4,7,8]. Circumferences were measured according to International Society for the Advancement of Kinanthropometry (ISAK) standards [12]. Mid-upper arm, mid-thigh, mid-calf, gluteal (hip), and waist circumferences were assessed on the right side to the nearest 0.1 cm using a non-elastic tape (KaWe CE; Kirchner und Wilhelm, Asperg, Germany). The waist-to-height ratio (WHtR) was calculated as waist circumference/height, and the waist-to-hip ratio (WHR) as waist circumference/hip circumference. All anthropometric and ultrasound measurements were performed by an ISAK Level 2–certified anthropometrist with >5 years of field experience. Pre-study intra-observer reliability testing demonstrated excellent repeatability (intraclass correlation coefficient > 0.95 for repeated measures).

### 2.3. Ultrasound Measurement Protocol

Ultrasound and anthropometric measurements were obtained at three timepoints: pre-race, after Stage 4, and after Stage 7. All measurement sites were located on the right side of the body and marked pre-race using a water-soluble cosmetic pencil (Schwan Cosmetics, Heroldsberg, Germany); markings were renewed as needed before subsequent measurements due to perspiration. Eight anatomical sites were assessed to capture inter-individual variation in SAT distribution: (1) upper abdomen (UA), (2) lower abdomen (LA), (3) erector spinae (ES), (4) distal triceps (DT), (5) brachioradialis (BR), (6) lateral thigh (LT), (7) front thigh (FT), and (8) medial calf (MC). Marking and positioning followed standard postural orientations: UA, LA, and LT in standing; DT and BR with the arm supported; FT and MC with the leg supported [13]. Ultrasound scans were conducted with participants in supine, prone, or rotated positions as appropriate. The protocol aligned with the International Olympic Committee Working Group recommendations for body-composition assessment [13]. Measurements were performed using a GE Logiq-e ultrasound system (GE Healthcare Austria, Vienna, Austria) with a linear transducer. To minimize tissue compression, a thick layer of ultrasound gel was applied, and the probe was placed with only minimal pressure on the skin [13]. Images were analyzed using FAT software (FAT 3.2; Rotosport, Stattegg, Austria), designed for semi-automated SAT quantification, with a constant sound speed of 1450 m/s for calibration [14]. An automated distance algorithm was used, with manual correction when necessary [8,13,15]. The software quantified SAT thickness including (I) and excluding (E) embedded fibrous structures (F). For each site, the mean value within the region of interest was recorded as the measurement. The sum of the eight sites yielded D_I_ (including fibrous structures) and D_E_ (excluding fibrous structures). The fibrous component was calculated as D_F_ = D_I_ − D_E_, and expressed as a percentage: D_F,%_ = 100 × D_F/_D_I_. All ultrasound assessments were performed by the same trained observer at each time point to minimize inter-observer variability.

### 2.4. Statistical Analysis

All analyses were conducted using MINITAB 17.2 (Minitab, Inc., State College, PA, USA). Descriptive data are reported as mean ± SD. Normality was assessed with the Kolmogorov–Smirnov test. As most variables met normality assumptions (*p* > 0.05), parametric tests were applied for consistency across outcomes. Paired Student’s t-tests were used to compare prerace vs. Stage 4 and prerace vs. Stage 7 values for BM, BMI, MI_I_, WHR, WHtR; site-specific SAT thickness (UA, LA, ES, DT, BR, LT, FT, MC) including and excluding fibrous structures; total D_I_ and D_E_; and D_F_ components. Associations among variables were examined using Spearman’s correlation coefficient, chosen for its reduced sensitivity to outliers. Statistical significance was defined as *p* < 0.05.

## 3. Results

Environmental and elevation characteristics for each stage are presented in Table 1.

Baseline characteristics of the ultrarunners are summarized in Table 2.

### 3.1. Anthropometric and SAT Thickness Parameters and Their Changes During the Race

Baseline anthropometric variables (BM, BMI, MI_I_, WHR, WHtR) and their values after Stage 4 (A4) and Stage 7 (A7) are shown in Table 3.

The distribution of SAT excluding embedded fibrous structures at prerace, mid-race (after Stage 4), and post-race (after Stage 7) is shown in Figure 1. After Stage 4, significant absolute decreases were observed in the sum of D_E_ (*p* = 0.001) and the sum of D_I_ (*p* = 0.006), as well as at specific sites: UA_E_ (*p* = 0.035), LA_E_ (*p* = 0.001), LT_E_ (*p* = 0.026), LT_I_ (*p* = 0.028), and LA_I_ (*p* = 0.002). Following Stage 7, further significant reductions were found in total D_E_ (*p* < 0.001) and D_I_ (*p* = 0.001), and at individual sites: LA_E_ (*p* = 0.001), LT_E_ (*p* = 0.014), FT_E_ (*p* = 0.025), LA_I_ (*p* = 0.004), LT_I_ (*p* = 0.011), and FT_I_ (*p* = 0.041).

Regarding associations between total SAT including and excluding fibrous structures, changes in D_E_ from prerace to postrace (A7–B1) were significantly correlated with changes in LA_E_ (*r* = 0.71, *p* < 0.001), UA_E_ (*r* = 0.47, *p* = 0.037), and DT_E_ (*r* = 0.52, *p* = 0.020). Similarly, changes in D_I_ were significantly associated with changes in LA_I_ (*r* = 0.79, *p* < 0.001) and LT_I_ (*r* = 0.68, *p* = 0.001) over the same interval.

### 3.2. Individual Anthropometric Profiles and SAT Changes During the Race

At prerace, four male ultrarunners presented with BMI ≥ 25 kg·m^−2^; in one of these, WHR exceeded 0.90 and WHtR exceeded 0.50. After Stage 4, three athletes still exhibited BMI ≥ 25 kg·m^−2^, with one maintaining WHR > 0.90 and a D_E_ of 56.9 mm. By post-race (after Stage 7), BMI ≥ 25 kg·m^−2^ persisted in two runners. Individual trajectories are illustrated in Figure 2.

In these athletes, the corresponding sums of SAT excluding embedded fibrous structures (D_E_) ranged from 34.6 to 64.4 mm. After Stage 7, D_E_ values ranged from 31.4 to 49.6 mm. A WHR > 0.90 was again observed in one runner, accompanied by a D_E_ of 53.6 mm. Individual SAT trajectories are presented in Figure 3.

Significant associations between prerace anthropometric indices and SAT distribution are summarized in Table 4. The strongest prerace associations were observed for BMI, followed by MI_I_, WHtR, and BM, primarily with abdominal SAT measurements (UA_E_, UA_I_ and LA_E_). WHR showed no significant relationships with any SAT parameter and is therefore not included in the table.

Significant associations between anthropometric indices and SAT distribution at mid-race (A4) are summarized in Table 5. The strongest relationships were observed for BMI and MI_I_, followed by WHtR. BM showed no significant association with any SAT site. As at prerace, WHR did not correlate significantly with SAT parameters and is therefore not included.

Significant post-race associations between anthropometric indices and SAT distribution are summarized in Table 6. As at earlier timepoints, WHR showed no significant relationships with SAT parameters and is therefore not included.

The strongest associations were observed for BMI, followed by WHtR and MI_I_. Body mass showed no significant correlation with any SAT site (Figure 4).

### 3.3. Embedded Fibrous Structures and Their Changes During the Race

The distribution of embedded fibrous structures at prerace and their changes during the race are summarized in Table 7.

Prerace, the percentage of embedded fibrous structures (D_F,%_) was strongly and inversely associated with total SAT thickness, both excluding and including fibrous structures (*r* = −0.93, *p* < 0.001 for both). D_F,%_ was also significantly negatively correlated with BMI, MI_I_, and WHtR (*r* = −0.53, *p* = 0.015; *r* = −0.57, *p* = 0.009; *r* = −0.59, *p* = 0.006, respectively). Across the race, D_F,%_ remained significantly inversely related to D_E_ (*r* = −0.67, *p* = 0.002).

Post-race (after Stage 7), significant increases in fibrous content were observed at the upper abdomen (F_UA,%_, *p* = 0.022), lower abdomen (F_LA,%,_ *p* = 0.014), erector spinae (F_ES,%_, *p* = 0.038), and in overall D_F,%_ (*p* = 0.001) (Table 7). At this timepoint, D_F,%_ was significantly negatively associated with BMI (*r* = −0.50, *p* = 0.024), MI_I_ (*r* = −0.48, *p* = 0.034), WHtR (*r* = −0.48, *p* = 0.031), D_E_ (*r* = −0.94, *p* < 0.001), and D_I_ (*r* = −0.90, *p* < 0.001). Changes in F_LA,%_ were inversely correlated with changes in LA_E_ (∆F_LA,%_ vs. ∆LA_E_: *r* = −0.64, *p* = 0.003).

## 4. Discussion

The principal finding of this study is that BMI, MI_I_ and WHtR were more strongly associated with ultrasound-derived SAT distribution than BM alone. These indices consistently correlated with total and regional SAT before, during, and after a 7-day ultramarathon, suggesting that they provide more informative indicators of adiposity and compositional change under real-world multistage race conditions.

### 4.1. Principal Findings in Context and Advancement of Knowledge

Across pre-, mid-, and post-race assessments, ultrasound-derived SAT decreased significantly, both when excluding and including embedded fibrous structures. The most pronounced site-specific reductions occurred at the abdomen and thighs–most consistently the lower and upper abdomen and the lateral thigh, and by race end also the front thigh–corresponding to regions with the highest prerace SAT thickness. This distribution aligns with previous ultrasound studies identifying the upper and lower abdomen and lateral thigh as key depots of subcutaneous fat [10,16,17]. Although BM declined over the race, it showed no significant association with site-specific SAT at any time point.

In contrast, BMI, MI_I_, and WHtR closely tracked both total and regional SAT. At all timepoints, BMI and MI_I_ were associated with D_E_, D_I_, and abdominal and erector spinae SAT; WHtR showed similar relationships, whereas WHR demonstrated no significant association with SAT at any site and was elevated only in a single outlier. Taken together, these findings indicate that BMI, MI_I_, and WHtR outperform BM as proxies for adiposity distribution in the context of a multistage ultramarathon.

These observations are consistent with broader evidence that WHtR can outperform BMI and waist circumference in capturing central fat distribution and cardiometabolic risk, while BMI alone has recognized limitations for body-composition assessment [1,5,18,19,20,21]. In this small, non-professional cohort, BMI did not clearly underperform MI_I_, in contrast to previous reports [2]; differences in sample size, age range, and training status may contribute.

This study extends existing knowledge by implementing repeated, field-based, site-specific ultrasound assessments of SAT during a multistage ultramarathon and by incorporating microstructural characterization of embedded fibrous structures. The robust inverse relationships observed between the percentage of embedded fibrous structures and SAT thickness (particularly at the lower abdomen) across race stages are compatible with the hypothesis that sustained mechanical loading during multistage ultra-endurance exercise is accompanied by structural adaptations within SAT [2,11].

### 4.2. Practical Implications for Athletes, Coaches, and Support Teams

Monitoring: For routine field monitoring during multistage ultramarathons, BMI, MI_I_, and WHtR provide more informative indicators of adiposity distribution than BM alone. WHR is not recommended as a primary index in this context.

Priority sites: When ultrasound is available, emphasis should be placed on the lower and upper abdomen and the thighs, which exhibit the earliest and most pronounced SAT changes and allow for efficient, site-sparse assessments.

Decision-making: Reductions in BM should not be interpreted as equivalent to fat loss. Combining simple anthropometric indices with site-specific ultrasound-derived SAT and basic hydration assessment offers a more appropriate basis for decisions on fueling and recovery between stages.

Risk management: Persistently elevated abdominal SAT in combination with higher BMI, MI_I_, or WHtR may help identify athletes who could benefit from individualized body-composition management during preparation for multistage events.

### 4.3. Strengths and Limitations

Key strengths of this study include repeated, ecologically valid assessments at pre-, mid-, and post-race; site-specific ultrasound measurements of SAT with differentiation of embedded fibrous structures; and the combined evaluation of BMI, MI_I_, and WHtR against BM to identify simple, field-feasible proxies. Intra-observer reliability was high (ICC > 0.95).

A primary limitation is that direct hydration markers (e.g., urine specific gravity/osmolality, bioimpedance-derived total body water, plasma osmolality) were not included in the present analyses. Consequently, short-term changes in BM cannot be clearly separated from fluid shifts. These hydration data were collected within the broader research program on this cohort and are reported elsewhere [22]; integrating them here would extend beyond the scope of this article, which specifically focuses on ultrasound-derived SAT and anthropometric indices.

Generalizability is further constrained by the small, predominantly male sample, the limited age range, and the focus on a single multistage event in non-professional runners. For embedded fibrous structures, normative data and construct validation are still emerging; related interpretations should therefore be regarded as preliminary and hypothesis-generating [2,11]. Finally, anthropometric indices–although useful–remain proxy measures and do not replace criterion body-composition methods.

### 4.4. Directions for Future Research

Future research should, firstly, integrate hydration endpoints collected in this cohort (e.g., urine specific gravity/osmolality, bioimpedance-derived total body water, plasma osmolality) with ultrasound-derived SAT measures (D_E_, D_I_, D_F_, D_F,%_) and anthropometric indices (BMI, MI_I_, WHtR) across pre-, mid-, and post-race time points to distinguish fluid-related from tissue-related changes, and, where appropriate, link these analyses with the companion paper on hydration outcomes [22]. Secondly, ultrasound-derived measures (D_E_, D_I_, D_F_, D_F,%_) should be benchmarked against criterion methods such as DXA and MRI to quantify site-specific absolute and relative error, derive calibration equations, and establish minimal detectable changes. Thirdly, larger, sex-balanced cohorts from multiple multistage events should be enrolled, with analyses stratified by sex, training status, and age to enhance generalizability and explore potential sex × site interactions. Fourthly, predictive models ought to be developed and preregistered that combine prerace morphology, simple in-race indices (BMI, MI_I_, WHtR), and site-specific SAT measures to forecast within-event adipose changes and recovery demands. Fifthly, direct comparisons of BMI and MI_I_, and evaluation of whether WHtR consistently outperforms both for tracking regional SAT in multistage ultramarathons, should include discrimination, calibration, and reclassification metrics [7,10,16,17,23,24]. Finally, normative datasets for embedded fibrous structures (D_F_, D_F,%_ by site) should be expanded across sports, sexes, and age groups, with assessment of temporal dynamics, relationships to performance and recovery, and formal evaluation of construct validity and inter- and intra-rater reliability to support standardized reporting.

## 5. Conclusions

In this 7-day multistage ultramarathon, BMI, MI_I_, and WHtR–but not BM–closely reflected ultrasound-derived SAT distribution across pre-, mid-, and post-race assessments. The largest SAT reductions occurred at the abdomen and thighs, while WHR showed no meaningful associations apart from a single outlier.

Given the small, predominantly male sample and the absence of integrated hydration analysis, these findings should be regarded as preliminary. The combined use of site-specific ultrasound and simple anthropometric indices appears promising for field monitoring of body composition in multistage ultrarunners, but confirmation in larger, sex-balanced cohorts with standardized hydration assessment is needed before definitive recommendations can be made.

## Figures and Tables

**Figure 1 jfmk-10-00467-f001:**
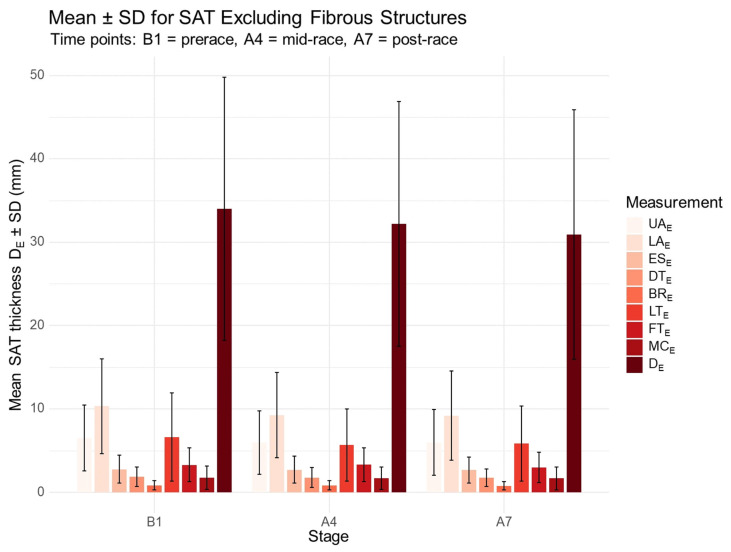
Distribution of SAT thickness excluding fibrous structures at prerace and during the race. B1, prerace; A4, after Stage 4; A7, after Stage 7. UA, upper abdomen; LA, lower abdomen; ES, erector spinae; DT, distal triceps; BR, brachioradialis; LT, lateral thigh; FT, front thigh; MC, medial calf; E, excluding fibrous structures; D_E_, sum of SAT thicknesses at the eight sites excluding embedded fibrous structures.

**Figure 2 jfmk-10-00467-f002:**
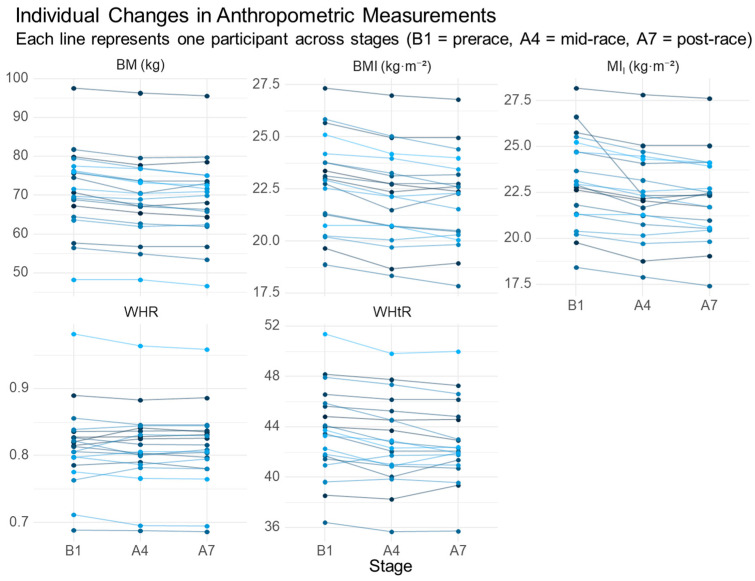
Individual changes in anthropometric indices across the race. B1, prerace; A4, after Stage 4; A7, after Stage 7. BM, body mass; BMI, body mass index; MI_I_, improved mass index; WHR, waist-to-hip ratio; WHtR, waist-to-height ratio. Each line represents one participant across stages (B1 = prerace, A4 = mid-race, A7 = post-race).

**Figure 3 jfmk-10-00467-f003:**
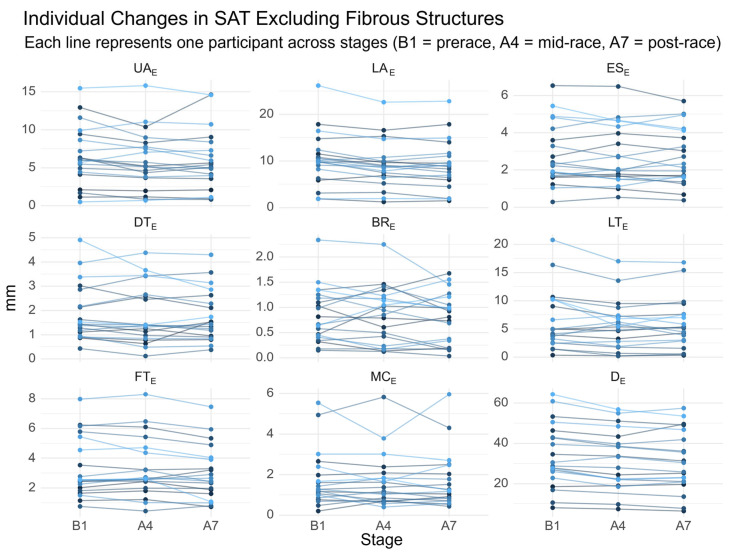
Individual changes in SAT thickness excluding fibrous structures. UA, upper abdomen; LA, lower abdomen; ES, erector spinae; DT, distal triceps; BR, brachioradialis; LT, lateral thigh; FT, front thigh; MC, medial calf; e, excluding fibrous structures; D_e_, sum of SAT thicknesses at the eight sites excluding embedded fibrous structures. Each line represents one participant across stages (B1 = prerace, A4 = mid-race, A7 = post-race).

**Figure 4 jfmk-10-00467-f004:**
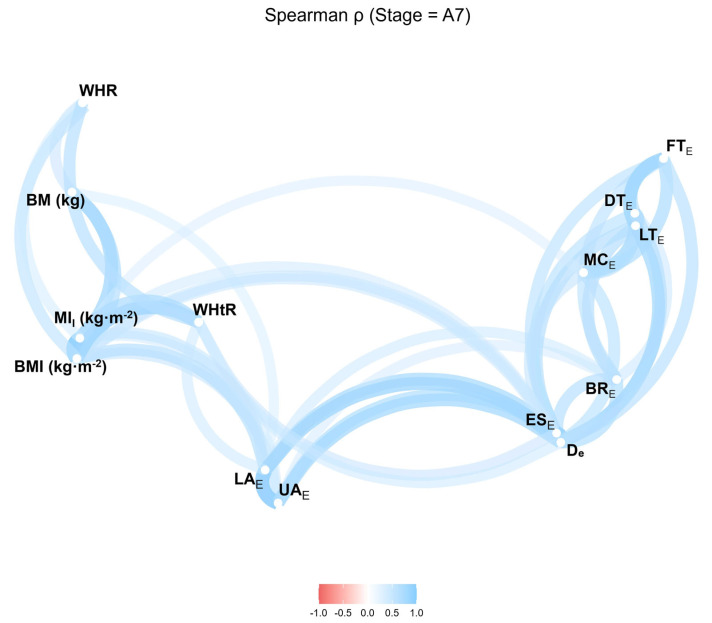
Correlation network of anthropometric indices and SAT thicknesses excluding fibrous structures after Stage 7. BMI, body mass index; MI_I_, improved mass index; WHR, waist-to-hip ratio; WHtR, waist-to-height ratio; UA, upper abdomen; LA, lower abdomen; ES, erector spinae; DT, distal triceps; BR, brachioradialis; LT, lateral thigh; FT, front thigh; MC, medial calf. Values range from −1.0 to 1.0 (Spearman’s *r*).

**Table 1 jfmk-10-00467-t001:** Stages of the Moravian Ultra Marathon: elevation profile and general weather conditions.

Stage	Ascent (m)	Descent (m)	General Weather Conditions	Mean Temperature (°C)	Humidity (%)
1	830	830	sunny, partly cloudy	22.1	76
2	830	862	sunny	24.3	70
3	830	734	sunny, partly cloudy	24.8	74
4	900	787	sunny	30.2	80
5	760	948	sunny	23.1	78
6	911	1055	sunny, partly cloudy	19.9	74
7	740	740	sunny	18.6	70

**Table 2 jfmk-10-00467-t002:** Baseline characteristics and anthropometric data of ultrarunners.

Characteristics	Mean ± SD
Age (years)	43.1 ± 7.0
Years of active running (years)	13.5 ± 11.3
Training hours per week *	6.7 ± 2.7
Body height (m)	1.8 ± 0.1
Sitting height (m)	0.9 ± 0.0

* Training volume during the three months preceding the competition.

**Table 3 jfmk-10-00467-t003:** Anthropometric variables at prerace, after Stage 4 (A4), and after Stage 7 (A7).

Characteristics	Prerace	A4 (Mid-Race)	A7 (Post-Race)
Body mass (kg)	71.3 ± 10.6	69.5 ± 10.3 **	69.1 ± 10.5 **
BMI (kg·m^−2^)	23.0 ± 2.4	22.2 ± 2.2 **	22.0 ± 2.2 **
MI_I_ (kg·m^−2^)	23.1 ± 2.5	22.3 ± 2.4 **	22.1 ± 2.3 **
WHR	0.813 ± 0.1	0.812 ± 0.1	0.810 ± 0.1
WHtR	0.436 ± 0.0	0.429 ± 0.0 **	0.428 ± 0.0 **

A4, after Stage 4; A7, after Stage 7; BMI, body mass index; MI_I_, improved mass index; WHR, waist-to-hip ratio; WHtR, waist-to-height ratio. Values are mean ± SD. ** *p* < 0.01 for comparisons between prerace and A4, and between prerace and A7.

**Table 4 jfmk-10-00467-t004:** Significant prerace associations between anthropometric indices and SAT distribution.

B1 (Prerace)			
BM	BMI	MI_I_	WHtR
BM vs. BMI (*r* = 0.56; *p* = 0.011)	x	x	x
BM vs. MI_I_ (*r* = 0.60; *p* = 0.005)	BMI vs. MI_I_ (*r* = 0.97; *p* < 0.001)	x	x
BM vs. WHR (*r* = 0.51; *p* = 0.021)	BMI vs. WHR (*r* = 0.56; *p* = 0.010)	MI_I_ vs. WHR (*r* = 0.48; *p* = 0.034)	x
x	BMI vs. WHtR (*r* = 0.87; *p* < 0.001)	MI_I_ vs. WHtR (*r* = 0.79; *p* < 0.001)	WHtR vs. WHR (*r* = 0.72; *p* < 0.001)
x	BMI vs. D_E_ (*r* = 0.52; *p* = 0.019)	MI_I_ vs. D_E_ (*r* = 0.54; *p* = 0.015)	WHtR vs. D_E_ (*r* = 0.54; *p* = 0.015)
x	BMI vs. D_I_ (*r* = 0.51; *p* = 0.023)	MI_I_ vs. D_I_ (*r* = 0.52; *p* = 0.018)	WHtR vs. D_I_ (*r* = 0.52; *p* = 0.018)
BM vs. UA_E_ (*r* = 0.47; *p* = 0.039)	BMI vs. UA_E_ (*r* = 0.56; *p* = 0.011)	MI_I_ vs. UA_E_ (*r* = 0.57; *p* = 0.009)	x
BM vs. UA_I_ (*r* = 0.48; *p* = 0.033)	BMI vs. UA_I_ (*r* = 0.57; *p* = 0.009)	MI_I_ vs. UA_I_ (*r* = 0.58; *p* = 0.007)	x
BM vs. LA_E_ (*r* = 0.50; *p* = 0.027)	BMI vs. LA_E_ (*r* = 0.61; *p* = 0.004)	MI_I_ vs. LA_E_ (*r* = 0.61; *p* = 0.003)	WHtR vs. LA_E_ (*r* = 0.50; *p* = 0.026)
x	BMI vs. LA_I_ (*r* = 0.61; *p* = 0.004)	MI_I_ vs. LA_I_ (*r* = 0.58; *p* = 0.007)	WHtR vs. LA_I_ (*r* = 0.52; *p* = 0.020)
x	BMI vs. ES_E_ (*r* = 0.54; *p* = 0.014)	MI_I_ vs. ES_E_ (*r* = 0.53; *p* = 0.017)	WHtR vs. ES_E_ (*r* = 0.51; *p* = 0.021)
x	BMI vs. ES_I_ (*r* = 0.49; *p* = 0.027)	MI_I_ vs. ES_I_ (*r* = 0.50; *p* = 0.026)	x

B1, prerace; BM, body mass; BMI, body mass index; MI_I_, improved mass index; WHR, waist-to-hip ratio; WHtR, waist-to-height ratio; D_E_, sum of SAT thicknesses at eight sites excluding fibrous structures; D_I_, sum of SAT thicknesses at eight sites including fibrous structures; UA, upper abdomen; LA, lower abdomen; ES, erector spinae; E, excluding fibrous structures; I, including fibrous structures.

**Table 5 jfmk-10-00467-t005:** Significant mid-race (A4) associations between anthropometric indices and SAT distribution.

A4 (Mid-Race)			
BM	BMI	MI_I_	WHtR
BM vs. BMI (*r* = 0.67; *p* = 0.001)	x	x	x
BM vs. MI_I_ (*r* = 0.72; *p* < 0.001)	BMI vs. MI_I_ (*r* = 0.97; *p* < 0.001)	x	x
x	x	x	x
x	BMI vs. WHtR (*r* = 0.88; *p* < 0.001)	MI_I_ vs. WHtR (*r* = 0.78; *p* < 0.001)	x
x	BMI vs. D_E_ (*r* = 0.50; *p* = 0.028)	MI_I_ vs. D_E_ (*r* = 0.56; *p* = 0.013)	WHtR vs. D_E_ (*r* = 0.48; *p* = 0.036)
x	BMI vs. D_I_ (*r* = 0.54; *p* = 0.016)	MI_I_ vs. D_I_ (*r* = 0.59; *p* = 0.008)	WHtR vs. D_I_ (*r* = 0.52; *p* = 0.023)
x	BMI vs. UA_E_ (*r* = 0.47; *p* = 0.036)	MI_I_ vs. UA_E_ (*r* = 0.48; *p* = 0.032)	x
x	BMI vs. UA_I_ (*r* = 0.47; *p* = 0.038)	MI_I_ vs. UA_I_ (*r* = 0.49; *p* = 0.027)	x
x	BMI vs. LA_E_ (*r* = 0.68; *p* = 0.001)	MI_I_ vs. LA_E_ (*r* = 0.66; *p* = 0.002)	WHtR vs. LA_E_ (*r* = 0.67; *p* = 0.001)
x	BMI vs. LA_I_ (*r* = 0.67; *p* = 0.001)	MI_I_ vs. LA_I_ (*r* = 0.64; *p* = 0.002)	WHtR vs. LA_I_ (*r* = 0.66; *p* = 0.002)
x	BMI vs. ES_E_ (*r* = 0.56; *p* = 0.011)	MI_I_ vs. ES_E_ (*r* = 0.57; *p* = 0.009)	WHtR vs. ES_E_ (*r* = 0.58; *p* = 0.008)
x	BMI vs. ES_I_ (*r* = 0.56; *p* = 0.010)	MI_I_ vs. ES_I_ (*r* = 0.57; *p* = 0.008)	WHtR vs. ES_I_ (*r* = 0.59; *p* = 0.006)

A4, mid-race; BM, body mass; BMI, body mass index; MI_I_, improved mass index; WHR, waist-to-hip ratio; WHtR, waist-to-height ratio; D_E_, sum of SAT thicknesses at eight sites excluding fibrous structures; D_I_, sum of SAT thicknesses at eight sites including fibrous structures; UA, upper abdomen; LA, lower abdomen; ES, erector spinae; E, excluding fibrous structures; I, including fibrous structures.

**Table 6 jfmk-10-00467-t006:** Significant post-race associations between anthropometric indices and SAT distribution.

A7 (Post-Race)			
BM	BMI	MI_I_	WHtR
BM vs. BMI (*r* = 0.71; *p* < 0.001)	x	x	x
BM vs. MI_I_ (*r* = 0.77; *p* < 0.001)	BMI vs. MI_I_ (*r* = 0.94; *p* < 0.001)	x	x
x	x	x	x
x	BMI vs. WHtR (*r* = 0.82; *p* < 0.001)	MI_I_ vs. WHtR (*r* = 0.71; *p* < 0.001)	x
x	BMI vs. D_E_ (*r* = 0.48; *p* = 0.032)	MI_I_ vs. D_E_ (*r* = 0.47; *p* = 0.036)	WHtR vs. D_E_ (*r* = 0.49; *p* = 0.029)
x	x	x	WHtR vs. D_I_ (*r* = 0.48; *p* = 0.033)
x	BMI vs. UA_E_ (*r* = 0.52; *p* = 0.019)	MI_I_ vs. UA_E_ (*r* = 0.47; *p* = 0.037)	x
x	BMI vs. UA_I_ (*r* = 0.54; *p* = 0.014)	MI_I_ vs. UA_I_ (*r* = 0.50; *p* = 0.025)	x
x	BMI vs. LA_E_ (*r* = 0.63; *p* = 0.003)	MI_I_ vs. LA_E_ (*r* = 0.55; *p* = 0.012)	WHtR vs. LA_E_ (*r* = 0.50; *p* = 0.027)
x	BMI vs. LA_I_ (*r* = 0.60; *p* = 0.005)	MI_I_ vs. LA_I_ (*r* = 0.50; *p* = 0.026)	WHtR vs. LA_I_ (*r* = 0.48; *p* = 0.031)
x	BMI vs. ES_E_ (*r* = 0.49; *p* = 0.030)	x	WHtR vs. ES_E_ (*r* = 0.52; *p* = 0.020)
x	BMI vs. ES_I_ (*r* = 0.48; *p* = 0.031)	x	WHtR vs. ES_I_ (*r* = 0.47; *p* = 0.038)

A7, post-race; BM, body mass; BMI, body mass index; MI_I_, improved mass index; WHtR, waist-to-height ratio; D_E_, sum of SAT thicknesses at eight sites excluding fibrous structures; D_I_, sum of SAT thicknesses at eight sites including fibrous structures; UA, upper abdomen; LA, lower abdomen; ES, erector spinae; E, excluding fibrous structures; I, including fibrous structures.

**Table 7 jfmk-10-00467-t007:** Percentage distribution of embedded fibrous structures at prerace (B1), after Stage 4 (A4), and after Stage 7 (A7). Values are mean ± SD; * *p* < 0.05, ** *p* < 0.01 for comparisons between B1 vs. A4 and B1 vs. A7.

	Total Number		
	B1	A4	A7
F_UA,%_	14.7 ± 7.3	16.0 ± 9.0	18.6 ± 10.0 *
F_LA,%_	17.2 ± 9.7	19.1 ± 7.2	21.1 ± 10.2 *
F_ES,%_	10.8 ± 11.3	11.9 ± 7.1	13.8 ± 10.9 *
F_DT,%_	31.8 ± 13.3	28.8 ± 16.3	33.6 ± 11.8
F_BR,%_	12.7 ± 10.2	11.8 ± 10.3	13.6 ± 14.0
F_FT,%_	25.9 ± 13.0	25.4 ± 9.5	22.8 ± 12.9
F_LT,%_	24.1 ± 11.9	23.3 ± 13.6	25.4 ± 14.5
F_MC,%_	16.2 ± 11.7	17.5 ± 11.0	16.8 ± 11.8
D_F (mm)_	6.7 ± 1.4	6.6 ± 2.4	7.0 ± 1.9
D_F,%_	18.6 ± 6.3	19.4 ± 5.4	21.1 ± 7.3 **

B1, prerace; A4, after Stage 4; A7, after Stage 7. F, site-specific percentage of embedded fibrous structures at each SAT site (UA, upper abdomen; LA, lower abdomen; ES, erector spinae; DT, distal triceps; BR, brachioradialis; FT, front thigh; LT, lateral thigh; MC, medial calf). D_F_ represents the absolute sum of embedded structures across all eight sites. D_F,%_ is the percentage of fibrous structures embedded in SAT, calculated as D_F,%_ = 100 × (D_I_ − D_E_)/D_I_. Values are mean ± SD. * *p* < 0.05, ** *p* < 0.01 for comparisons between B1 vs. A4 and B1 vs. A7.

## Data Availability

The raw data supporting the conclusions of this article will be made available by the authors on request.

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
