# Peer review of "Ultrasound-Based Assessment of Subcutaneous Adipose Tissue Changes During a 7-Day Ultramarathon: Association with Anthropometric Indices, Not Body Mass"

_jfmk, 2025, doi:10.3390/jfmk10040467_

Round 1

Reviewer 1 Report

Comments and Suggestions for Authors

I have reviewed the paper entitled: “Ultrasound-based assessment of subcutaneous adipose tissue changes during a 7-day ultramarathon: Association with anthropometric indices, Not body mass”.

This study was aimed to evaluate anthropometric variables such as body mass (BM), body mass index (BMI), mass index (MII), waist-to-hip ratio (WHR) and waist-to-height ratio (WHtR) for exercise-induced changes in body composition, specifically in relation to subcutaneous adipose tissue (SAT). Anthropometric and ultrasound measurements of subcutaneous adipose tissue were carried out excluding and including embedded fibrous structures, before the race, after stages 4 and 7 in a real-world ultramarathon setting.

It is the original research paper, well designed and conducted, the obtained results are interesting, with good contribution to the field. Only minor changes in the structure of some sentences and some typographical and grammatical corrections are needed. My suggestion is to accept for publication after minor revision.

There are some suggestions for the study:

The topic covered in this manuscript is interesting. The obtained results support the initial hypothesis that ultrasound-based assessment of subcutaneous adipose tissue changes are associated with anthropometric indices, not body mass during a 7-day ultramarathon. The authors do make significant contribution to this subject and point to the new relevant research directions. 

Abstract is too long. According to the Instructions for authors, it is limited to 250 words maximum. Authors should shorten the abstract in accordance with the technical requirements of the Journal.

Introduction provides relevant theoretical background, statement of the problem and proposed approach.

Section Materials and methods is sufficiently detailed described. Is the observer certified anthropometrist for ultrasound fat measurement? Authors should emphasize whether anthropometric and ultrasound measurements were performed by a trained person with sufficient experience.

The results are abundant, well presented in the tables and figures, with good contribution to the field. The authors should precisely define in Table number 3 what the abbreviation B1 represents for easier readability. In the line 291 the authors refer to table 11, which does not exist in the manuscript. The discussion interprets the findings in the line with obtained results and with those from the literature.

The authors should approach conclusions with caution. The conclusions stated by the authors requires more scientific facts and more research given the sample size, gender (mostly men), and age.

The style of references should be in accordance with the technical requirements of the JFMK.

Comments on the Quality of English Language

The quality of English language is relatively good. Redundancy should be avoided, as well as unclear and too long sentences that make manuscript difficult to read and understand. Changes in the structure of some sentences and some typographical and grammatical corrections are needed. In this sense, it could be useful to have the help of colleague who is more experienced in writing scientific papers.

Author Response

Comments and Suggestions for Authors

I have reviewed the paper entitled: “Ultrasound-based assessment of subcutaneous adipose tissue changes during a 7-day ultramarathon: Association with anthropometric indices, Not body mass”.

This study was aimed to evaluate anthropometric variables such as body mass (BM), body mass index (BMI), mass index (MII), waist-to-hip ratio (WHR) and waist-to-height ratio (WHtR) for exercise-induced changes in body composition, specifically in relation to subcutaneous adipose tissue (SAT). Anthropometric and ultrasound measurements of subcutaneous adipose tissue were carried out excluding and including embedded fibrous structures, before the race, after stages 4 and 7 in a real-world ultramarathon setting.

It is the original research paper, well designed and conducted, the obtained results are interesting, with good contribution to the field. Only minor changes in the structure of some sentences and some typographical and grammatical corrections are needed. My suggestion is to accept for publication after minor revision.

We are grateful for the constructive and encouraging feedback provided by both reviewers. We have carefully addressed each suggestion point-by-point, revising the manuscript to enhance its scientific clarity, methodological transparency, and linguistic precision. All changes are marked in the revised file.

There are some suggestions for the study:

The topic covered in this manuscript is interesting. The obtained results support the initial hypothesis that ultrasound-based assessment of subcutaneous adipose tissue changes are associated with anthropometric indices, not body mass during a 7-day ultramarathon. The authors do make significant contribution to this subject and point to the new relevant research directions. 

Abstract is too long. According to the Instructions for authors, it is limited to 250 words maximum. Authors should shorten the abstract in accordance with the technical requirements of the Journal.

We agree with the expert reviewer and have shortened the Abstract to approximately 245 words, aligning with the Journal’s limit. We also streamlined the flow to explicitly connect aim, methods, and principal findings and to include the hypothesis (p. 1, lines 13–33).

Introduction provides relevant theoretical background, statement of the problem and proposed approach.

Section Materials and methods is sufficiently detailed described. Is the observer certified anthropometrist for ultrasound fat measurement? Authors should emphasize whether anthropometric and ultrasound measurements were performed by a trained person with sufficient experience.

We concur with the expert reviewer and have added the requested information to the manuscript (p. 3, lines 116–119; p. 4, lines 144–145). All anthropometric and ultrasound assessments were conducted by an ISAK Level 2–certified anthropometrist with more than five years of field experience in ultrasound-based fat measurement. Pre-study intra-observer reliability testing demonstrated excellent repeatability (ICC > 0.95 for repeated measures), now reported in Methods (p. 3, lines 1116–119). The certified anthropometrist referenced above, Marietta Sengeis (Leitung Fachbereich Anthropometrie), has been added to the Acknowledgments. She holds a PhD (Medical Sciences) from the Medical University of Graz with a focus on anthropometry and subcutaneous adipose tissue patterning in elite athletes using a novel ultrasound technique.

The results are abundant, well presented in the tables and figures, with good contribution to the field. The authors should precisely define in Table number 3 what the abbreviation B1 represents for easier readability. In the line 291 the authors refer to table 11, which does not exist in the manuscript.

We clarified the meaning of “B1” in Table 3 to enhance readability and corrected a mislabeled cross-reference.

The discussion interprets the findings in the line with obtained results and with those from the literature.

The authors should approach conclusions with caution. The conclusions stated by the authors requires more scientific facts and more research given the sample size, gender (mostly men), and age.

We concur with the reviewer’s call for caution in interpreting the results. Accordingly, we revised the Discussion and Conclusions to temper claims, present the findings with appropriate caveats, and acknowledge the study’s constraints – a small, predominantly male cohort with a limited age range (pp. 13, lines 339–344). The principal messages are retained but explicitly framed as provisional. As requested, we also added an explicit limitation to the Methods (p. 3, lines 93–96).

The style of references should be in accordance with the technical requirements of the JFMK.

In line with the reviewer’s comment, we meticulously checked and corrected the reference list to conform to JFMK’s formatting and technical guidelines.

Comments on the Quality of English Language

The quality of English language is relatively good. Redundancy should be avoided, as well as unclear and too long sentences that make manuscript difficult to read and understand. Changes in the structure of some sentences and some typographical and grammatical corrections are needed. In this sense, it could be useful to have the help of colleague who is more experienced in writing scientific papers.

We agree with the reviewer and have comprehensively revised the manuscript to enhance clarity and readability. The text underwent thorough language editing, correcting typographical and grammatical errors. In line with the reviewer’s suggestion, we shortened long, complex sentences and removed redundant wording to improve flow and comprehension.

Reviewer 2 Report

Comments and Suggestions for Authors

The study is conducted on a very interesting topic related to a poorly studied athlete population. The methodology of the study is properly conducted and results are properly reported. Substantial improvements are in any needed to improve the quality of the manuscript:

The keywords should differs from those which appear in the title to increase to chance to find this paper if published with a broader search strategy.

The reporting of the aim of the study is too long, it should be mentioned briefly and following the aim, a hypothesis of potential expected results is needed.

Does the study received a IRB approval? This is mandatory to be reported as number of protocol of approval as well as the exact day, month and year of approval.

It is unusual to report tables within the method section. It should be moved to supplementary materials

The discussion should be organised differently to improve its quality, it is fine to be presented as subsections however restructured as follows:

  • The main findings of this study and their comparison with previously published literature on the same topic, and how this help in the advancement of knowledge
  • The practical implication of these findings on athletes, coaches and teams
  • The strengths and limitations of the study, foremost the absence of an evaluation of hydration levels that reports the changes of the later before and after the competition
  • The new directions for future research needed on this topic

The conclusion section should be more focused to go direct to the point on the findings of the study.  

Author Response

Comments and Suggestions for Authors

The study is conducted on a very interesting topic related to a poorly studied athlete population. The methodology of the study is properly conducted and results are properly reported. Substantial improvements are in any needed to improve the quality of the manuscript:

We are grateful for the constructive and encouraging feedback provided by both reviewers. We have carefully addressed each suggestion point-by-point, revising the manuscript to enhance its scientific clarity, methodological transparency, and linguistic precision. All changes are marked in the revised file.

The keywords should differs from those which appear in the title to increase to chance to find this paper if published with a broader search strategy.

We agree with the reviewer’s recommendation and have expanded the keyword list accordingly (p. 1, lines 34–35).

The reporting of the aim of the study is too long, it should be mentioned briefly and following the aim, a hypothesis of potential expected results is needed.

In line with the reviewer’s comments, we shortened the aim to enhance scientific flow and clarity, made the hypothesis explicit at the end of the Introduction, and included a rationale for the methodological novelty – ultrasound-based differentiation of fibrous and non-fibrous SAT (p. 2, lines 63–71).

Does the study received a IRB approval? This is mandatory to be reported as number of protocol of approval as well as the exact day, month and year of approval.

We concur with the reviewer and confirm that the study received IRB approval; this information has been added to the opening paragraph of Materials and Methods (p. 2, lines 74–77).

It is unusual to report tables within the method section. It should be moved to supplementary materials

We agree with the reviewer’s recommendation and relocated Table 1 to the Results section (p. 5, lines 157–159), which is suitable given that it characterizes the plant/facility.

The discussion should be organised differently to improve its quality, it is fine to be presented as subsections however restructured as follows:

  • The main findings of this study and their comparison with previously published literature on the same topic, and how this help in the advancement of knowledge
  • The practical implication of these findings on athletes, coaches and teams
  • The strengths and limitations of the study, foremost the absence of an evaluation of hydration levels that reports the changes of the later before and after the competition
  • The new directions for future research needed on this topic

We appreciate the reviewer’s detailed guidance on restructuring the Discussion. We have fully revised and reorganized it into the suggested subsections to maximize clarity and scientific impact. We also thank the reviewer for highlighting the role of hydration. We agree that fluid shifts can influence the interpretation of body mass changes. Although direct hydration markers were collected within the same research program, they were not incorporated into the present analysis by design, to maintain focus on ultrasound-derived SAT and anthropometric indices; detailed hydration findings for this cohort have been reported elsewhere (Chlíbková D, Nikolaidis PT, Rosemann T, Knechtle B, Bednář J. Fluid Metabolism in Athletes Running Seven Marathons in Seven Consecutive Days. Front Physiol. 2018 Feb 12;9:91). We have updated the Limitations and Directions for Future Research to emphasize a planned cross-analysis with these hydration endpoints.

The conclusion section should be more focused to go direct to the point on the findings of the study.  

We agree with the reviewer’s suggestion. We have refocused and streamlined the Conclusions to directly highlight the study’s principal findings and implications.

Round 2

Reviewer 2 Report

Comments and Suggestions for Authors

Thankful to the authors for being responsive. Still the only requested change is in the Discussion section, in the Directions for future research subsection, numeration should be removed because it creat confusion and should appear as on paragraph, as authors are invited to report the information, as firstly…, secondly…, thirdly…, fourthly…, and finally. 

Author Response

Reviewer 2

Thankful to the authors for being responsive. Still the only requested change is in the Discussion section, in the Directions for future research subsection, numeration should be removed because it creat confusion and should appear as on paragraph, as authors are invited to report the information, as firstly…, secondly…, thirdly…, fourthly…, and finally. 

We thank the Reviewer for this helpful suggestion. The numbering in the “Directions for Future Research” subsection has been removed, and the content is now presented as a single paragraph structured with “firstly…finally,” as requested.